# Systematic selection between age and household structure for models aimed at emerging epidemic predictions

Lorenzo Pellis[1,2,3 ✉], Simon Cauchemez[4], Neil M. Ferguson [iD] [3] & Christophe Fraser [iD] [5]

Numerous epidemic models have been developed to capture aspects of human contact patterns, making model selection challenging when they fit (often-scarce) early epidemic data equally well but differ in predictions. Here we consider the invasion of a novel directly transmissible infection and perform an extensive, systematic and transparent comparison of models with explicit age and/or household structure, to determine the accuracy loss in predictions in the absence of interventions when ignoring either or both social components. We conclude that, with heterogeneous and assortative contact patterns relevant to respiratory infections, the model's age stratification is crucial for accurate predictions. Conversely, the household structure is only needed if transmission is highly concentrated in households, as suggested by an empirical but robust rule of thumb based on household secondary attack rate. This work serves as a template to guide the simplicity/accuracy trade-off in designing models aimed at initial, rapid assessment of potential epidemic severity.

[1] Department of Mathematics, University of Manchester, Manchester, UK. [2] Zeeman Institute and Warwick Mathematics Institute, University of Warwick, Warwick, UK. [3] MRC Centre for Global Infectious Disease Analysis, J-IDEA, School of Public Health, Imperial College, London, UK. [4] Mathematical Modelling of Infectious Diseases Unit, Institut Pasteur, UMR2000, CNRS, 75015 Paris, France. [5] Oxford Big Data Institute, Li Ka Shing Centre for Health Information and Discovery, Nuffield Department of Medicine, University of Oxford, Oxford, UK. ✉email: lorenzo.pellis@manchester.ac.uk

Patterns of interactions and contacts between individuals in populations are recognised as important factors shaping the spread of infectious diseases[1,2]. Mathematical models are routinely used in analysis of epidemic data and prediction of epidemic trends[3,4]. A wealth of different options for how to describe social interactions in mathematical models is available, ranging from assuming homogeneous mixing[5,6], through analytically tractable models that focus on households, age structure or other idealised properties of the contact network[1,2,7,8], to complex individual-based stochastic simulations[1,9–12]. However, few comparative tools exist to guide the initial choice of structure when designing a mathematical modelling study. One common approach is statistical, comparing models by their ability to fit the data, or seeking a trade-off between the quality of data fitting and model parsimony. However, when data are limited, like at the beginning of an emerging epidemic, different models can fit the same data equally well and nevertheless lead to different predictions. These methods would then favour the simplest model, and would therefore not highlight whether further data should be collected to parameterise a more complicated model whose predictions can be significantly different.

Another comparison approach, rather than ranking models or selecting the best one, retains them all and combines their predictions to quantify uncertainty not only owing to stochastic dynamics and unknown parameters/initial conditions[13,14], but also to lack of knowledge on the correct model structure. Such multi-model ensemble methods, well established in the field of numerical weather forecast[15], have recently started gaining momentum in epidemiology[16–18], with different modelling groups sometimes asked to perform predictions of the impact of realistic control policies based on shared scenarios[10,18–21]. These comparative studies have proven popular with complex models and policy makers concerned with over-reliance on advice from single modelling groups, but so far have provided little insight into the causes of differences between models. In addition, unidentifiability issues owing to the large number of parameters are sometimes not discussed, and often dealt with through an informal and difficult-to-reproduce calibration procedure[22]. Furthermore, comparing models by fitting them to the same set of data (or small number of shared scenarios[23]) provides results that are useful in that specific context but often hard to generalise to other settings[18,23] or pathogens.

Therefore, here we take a more mathematical approach and we compare a few popular and relatively simple models when varying widely and systematically the constraints they are required to fit (i.e., the hypothetical data we would observe and fit them to). Particular attention is devoted to being as transparent as possible in the details of the process and the unidentifiability issues encountered, and to understand how the different components of model structure interact and cause differences in predictions. In this sense, our approach is complementary to the multi-model ensemble studies discussed above.

To make extensive comparison computationally feasible, we focus on a specific context where numerous analytical results are available: assuming we are observing the early phase of an emerging epidemic of a novel, directly transmissible, human infection, we explore the joint contribution of age stratification and household structure on its spread in a large and fully susceptible population. There is an extensive literature on age stratification in epidemic modelling[6,24] and an increasing amount of empirical data on age-specific patterns of mixing[25–27]. There is an equally extensive literature on household structure[28–30] as well as empirical data on estimation of household transmission parameters[31–36]. Epidemic models with age and household structure are among the most analytically tractable, and are thus particularly suited to rapid use in an emerging epidemic situation.

Our aim is to compare four different models: an age- and household-stratified model (hereafter denoted AH), a pure age-stratified (A), a pure household-stratified (H) and an unstructured (U) homogeneously mixing model. We then test the ability of these different models to predict: the average final size ($z$), the average peak daily incidence ($\pi$) and the average time to the peak daily incidence ($t$) of a single epidemic wave. Given model AH includes all others as submodels, we compare predictions by assuming model AH perfectly represents reality and measuring other models' deviations from it. More specifically, we extensively explore the parameter space of model AH and for each parameter combination we compute, in addition to the outputs of interest, those quantities that are most likely to be measured or estimated from early data in the initial phase of the epidemic, namely: the basic reproduction number $R_0$, i.e., the average number of cases a typical case generates in a fully susceptible population (this is not trivial for models with households; see Methods and Supplementary Methods, Sections 1.2.4 and 1.4); the ratio of adults versus children among new incident cases; and the household secondary attack rate (SAR), i.e., the average size of a within-household outbreak. These quantities, hereafter referred to as the observables, are then used to map (i.e., compute deterministically, assuming no error or noise) the parameters of the models A, H and U, the outputs of which are then compared with the assumed-true ones of model AH and considered inaccurate when they differ by more than a specified threshold $\varepsilon$ in relative terms. In computing model outputs and observables, we assume permanent immunity following infection, as well as constant parameter values and no demographic changes, i.e., we have in mind a relatively fast infection with negligible disease-induced mortality and no intervention nor behavioural or environmental change. In addition, despite the models used here being stochastic, for maximum computational efficiency we only focus on average values (infinite population limit).

Despite the rather specific scenario and limited range of models, this work provides a simple rule of thumb that, early on in an outbreak of a new emerging infection, can help striking the crucial balance between model simplicity and prediction accuracy when choosing a model aimed at rapid initial assessment of likely epidemic severity in the absence of interventions.

## Results

**Overview.** As baseline scenario we consider: age and household structure of Great Britain, random mixing, $R_0 = 2$, adults and children equally infective (relative infectivity $\phi = 1$) and an accuracy threshold $\varepsilon = 5\%$. We fully explore the models' behaviour for a within-household infectious-adult-to-susceptible-adult transmission probability $p_{aa}$ ranging from 0 to 0.95 and for children from equally to four times as susceptible as adults (relative susceptibility $\psi$ ranging from 1 to 4) in Fig. 1. We then compare models in Fig. 2. Other values of $R_0$ (from 1.1 to 4) and $\phi$ (from 1 to 2), UK-like contact patterns relevant to respiratory infections and a markedly different social structure (corresponding to Sierra Leone) are then explored in Figs. 3 and 4. Further sensitivity analyses, including intermediate contact patterns and social structures, other thresholds $\varepsilon$ (1% and 10%), and the case of children less susceptible/infective than adults ($\phi, \psi < 1$), are reported in the Supplementary Discussion (Sections 2.3.1, 2.3.4, 2.1.3 and 2.3.3, respectively).

**Baseline scenario.** We first describe the outputs of each model in absolute terms in the baseline scenario. Model U predicts a final size $z$ of 80%, a peak daily incidence $\pi$ of 5.5% and $t = 9.8$ generation times to the peak (we chose the generation time[37] as the time unit to facilitate translation to other infections). These values are common to all models (Fig. 1, bottom-left corner of all panels)

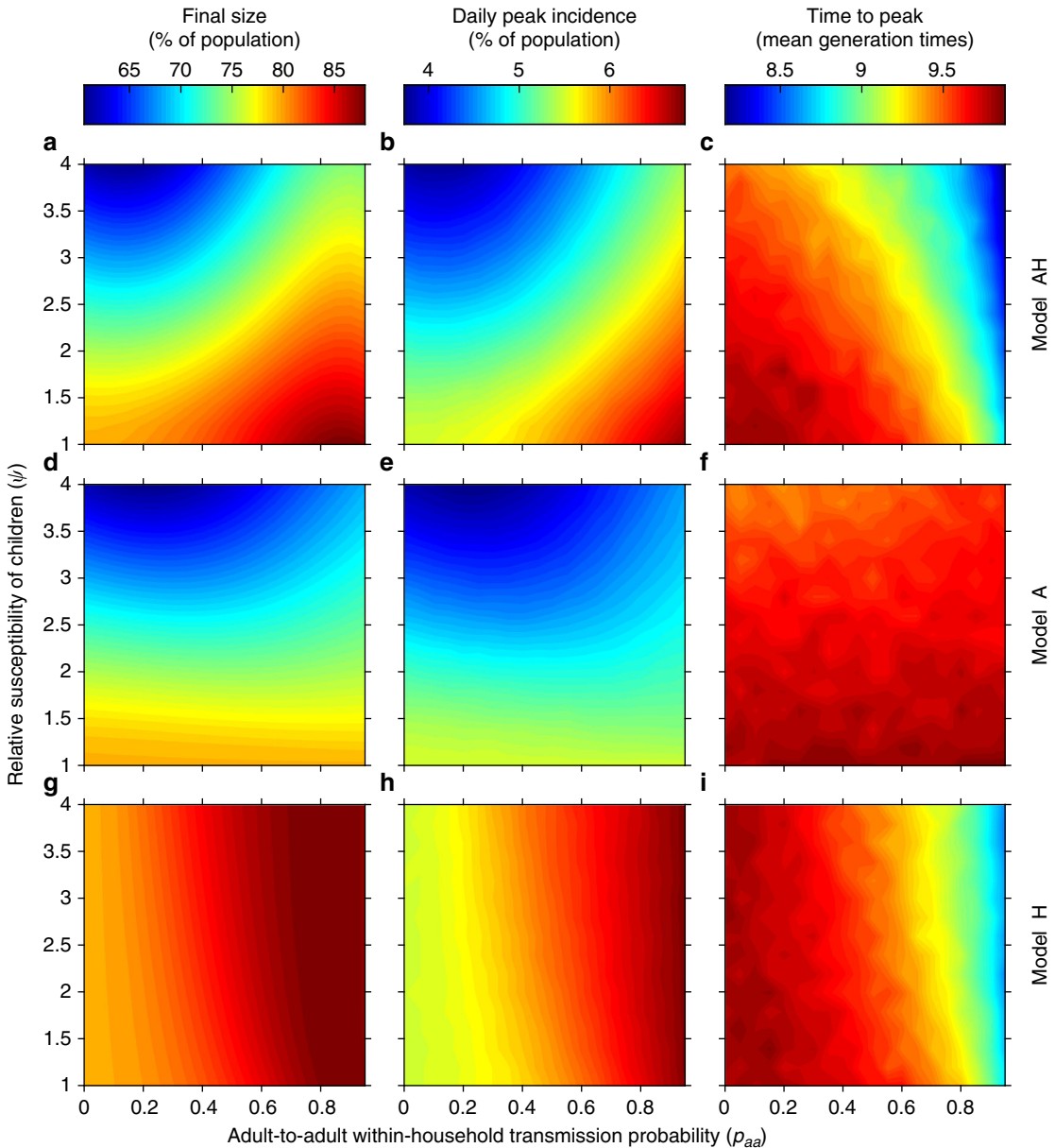

**Fig. 1 Models' output under baseline scenario.** For models AH (first row: **a**–**c**), A (second row: **d**–**f**) and H (third row: **g**–**i**), the average epidemic final size $z$ (first column: **a**, **d** and **g**), average peak daily incidence $\pi$ (second column: **b**, **e** and **h**) and average time to peak daily incidence $t$ (third column: **c**, **f** and **i**; time expressed in multiples of the generation time $T_G$) are plotted as functions of the within-household transmission probability $p_{aa}$ and the relative susceptibility of children versus adults $\psi$. The top row (model AH) is assumed to be a perfect representation of reality. The outputs of model U are not plotted, as they are independent of both variables on the axes (they depend only on $R_0$), but can be read in the bottom-left corner of panels in each column, where predictions of all models coincide. The baseline scenario assumes: population structure of Great Britain, random mixing (same contact rates for adults and children in all environments, $\gamma_g = \gamma_h = 1$, and global assortativity of children $\theta_g = 22.73\%$—within-household random mixing is always assumed), $R_0 = 2$ and children as infectious as adults ($\phi = 1$).

when there is no household transmission (within-household transmission probability $p_{aa} = 0$) and children and adults are equally susceptible (relative susceptibility $\psi = 1$). For a fair comparison, the parameter explorations for all models are performed at fixed $R_0$, i.e., larger values of the within-household transmission probability $p_{aa}$ correspond to suitably smaller values of between-household transmission. Both the final size $z$ (Fig. 1a) and the peak incidence $\pi$ (Fig. 1b) appear to decrease when increasing age difference in susceptibility (i.e., increasing $\psi$): this result is known for the final size in the absence of households[5,38,39], but appears to hold also for the peak incidence, and to extend to any amount of within-household transmission. Conversely, both outputs increase

when transmission is shifted from between to within households (increasing $p_{aa}$ at constant $R_0$). Comparing Fig. 1a, d and g and b, e and h, it appears that, for both final size $z$ and peak incidence $\pi$, model A is better than model H at mirroring the qualitative behaviour of model AH (though note that model H has only two free parameters, whereas model A has four, of which two shared with model AH: see description of model A in the Methods and Supplementary Discussion, Section 2.1.2). It also appears that the joint contribution of age and household structure in model AH can be roughly decomposed in the separate contribution of the two social structures (Fig. 1a, b contain elements of d and e for increasing $\psi$, and of g and h for increasing $p_{aa}$), but that models A

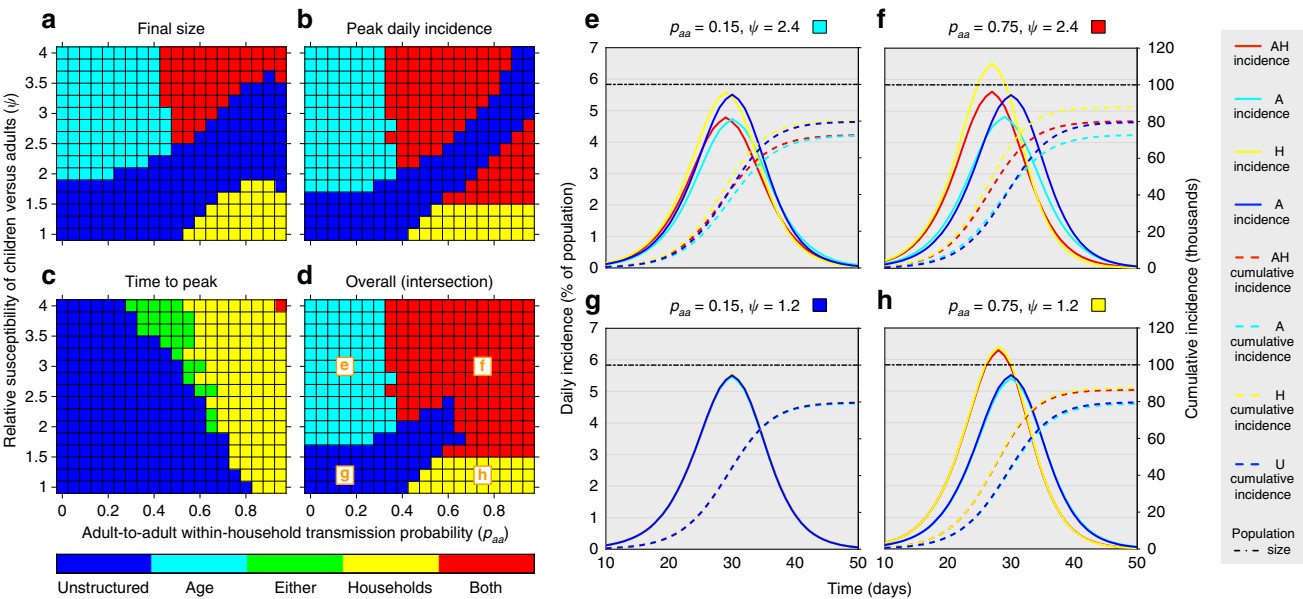

**Fig. 2 Simplest-model acceptance regions and examples of epidemic dynamics.** Colours refer to regions of parameter space where no (dark blue), each (age: light blue; households: yellow; either, indifferently: green) or both (red) forms of social structure need to be included in the model in order to compute within 5% relative accuracy of the assumed-true outputs of model AH: **a** the final size; **b** the peak daily incidence; **c** the time to peak daily incidence; and **d** all of them simultaneously (overall simplest-model acceptance regions plot, where simpler models are discarded if at least one of the three outputs is not sufficiently accurate). Parameters are as per baseline scenario (population structure of Great Britain, random mixing, $R_0 = 2$, $\phi = 1$). On the right-hand side **e–h**, epidemic dynamics of the four models at different parameter regimes indicated by the corresponding points labelled in orange in panel **d**: daily incidence (solid line, left axis) and cumulative incidence (dashed line, right axis) are plotted for models AH (red), A (light blue), H (yellow) and U (dark blue). The black dash-dotted line gives the total population size (right axis), to which cumulative incidence should be compared.

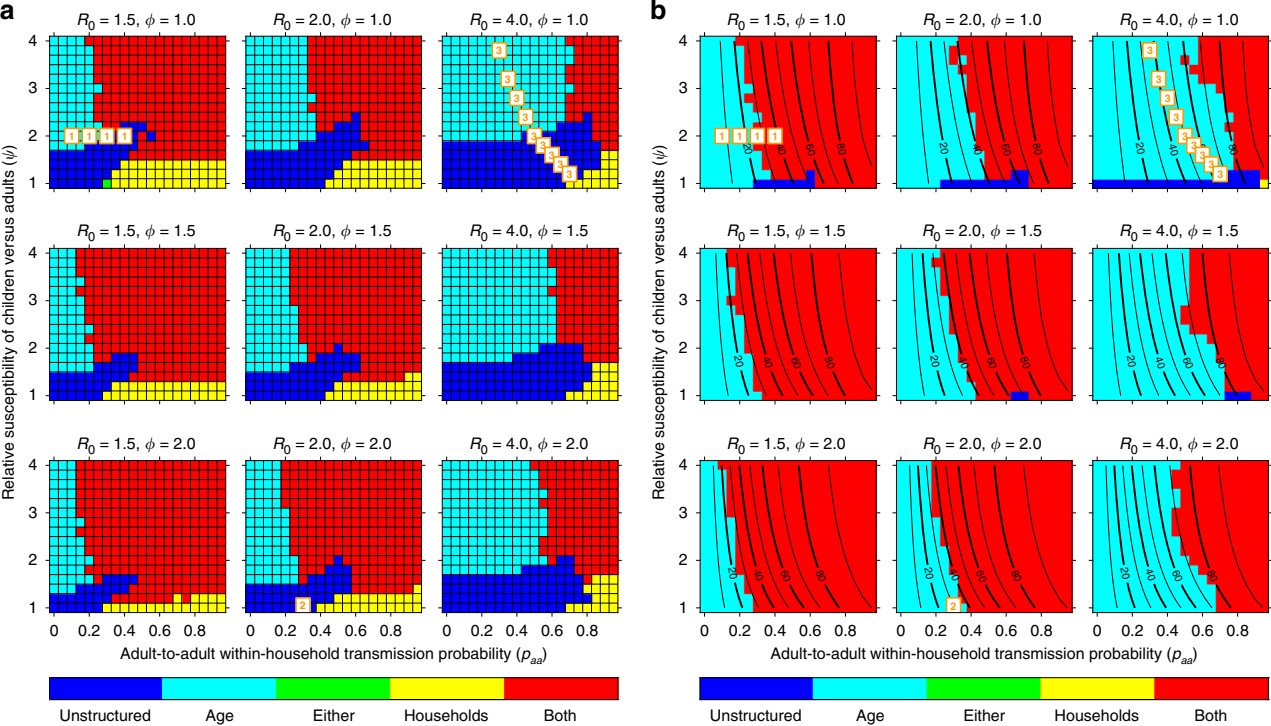

**Fig. 3 Sensitivity analysis on overall simplest-model acceptance regions. a** Overall simplest-model acceptance regions for various values of $R_0$ (1.5, 2 and 4; increasing from left to right) and the relative infectivity of children versus adults $\phi$ (1, 1.5 and 2; increasing from top to bottom) for the population structure of Great Britain, assuming baseline random mixing. Simple models are rejected when at least one of the three predicted outputs (final size, peak daily incidence and time to peak incidence) differs in relative terms by more than 5% from those of model AH (the panel corresponding to $R_0 = 2$ and $\phi = 1$ is identical to Fig. 2d). The overlaid orange labels cover the zones of the parameter space approximately associated to some realistic infections: (1) H1N1 2009 pandemic influenza; (2) 1918 pandemic influenza; (3) a highly infectious disease, with characteristics similar to measles or chickenpox. **b** Same plot as in **a**, but for UK-like contact patterns ($\gamma_g = \gamma_h = 0.75$) and assortative mixing ($\theta_g = 58\%$), with overlaid contour lines for the SAR (%, black lines).

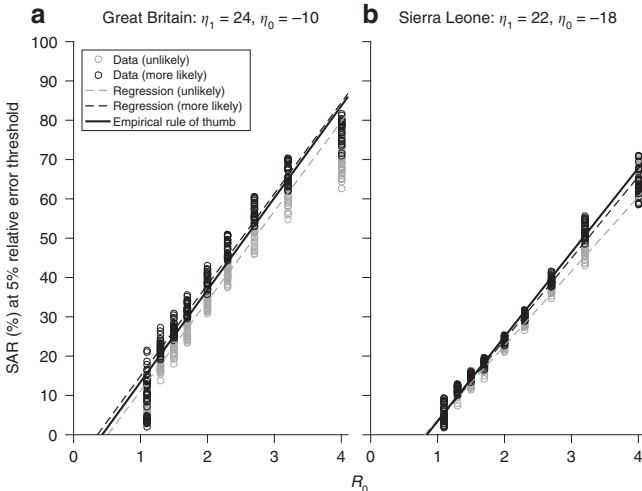

**Fig. 4 Criterion for inclusion of household structure for accurate predictions.** Empirical rule of thumb specifying the level of SAR above which a household structure is necessary for output predictions within a 5% relative accuracy for the household structure of **a** Great Britain (fraction of children = 22.73%, mean household size $\chi = 2.35$) and **b** Sierra Leone (fraction of children = 53.81%, mean household size $\chi = 5.85$), both with UK-like contact patterns and assortative mixing (though of little relevance, as the same rule of thumb also applies for random mixing: see Supplementary Discussion, Section 2.4). For each $R_0$, data points represent the values of the SAR along the borders between the light blue and red regions in Fig. 3b (more precisely, the values of the SAR at each point of the 5% relative error contour lines, which are not explicitly drawn here to reduce clutter, but are shown as think black lines in Supplementary Fig. 16B) for $\phi = 1$, 1.5 and 2 and $\psi$ ranging from 1 to 4. Black data points give SAR values associated with more moderate and realistic differences in susceptibility and infectivity between adults and children ($\psi\phi \leq 3$) and are typically less widespread than grey data points, which correspond to extreme and unrealistic differences ($\psi\phi > 3$). Regression lines (dashed) fit corresponding-colour data points, excluding $R_0 = 1.1$ and 4, where a linear fit looks unreasonable. The thick black line shows the empirical rule of thumb, with parameters reported in the figure ($\eta_1$, regression coefficient; $\eta_0$, intercept), and is empirically chosen to: be closer to the dark than the grey dashed line, reflecting low confidence in extreme differences between adults and children; and provide an acceptable fit for both the SAR values obtained assuming UK-like mixing (here) and random mixing (see Supplementary Discussion, Section 2.4). Note that part of the spread in the data points is owing to the ruggedness of the borders between regions (contour lines in Supplementary Fig. 16B): a smaller step in $p_{aa}$ and $\psi$ would reduce the spread but increase the computational cost.

and H, respectively, underestimate and overestimate both the final size and the peak incidence. Predictions of time to the peak $t$ (Fig. 1c) seem to decrease when increasing both $p_{aa}$ and $\psi$, but are more complex to interpret and overall show limited variation (much variation would come from stochastic delays in the very early epidemic phase, but these are minimised on purpose to detect structural differences in epidemic speed between models). Results are qualitatively similar for other values of $R_0$ and relative infectivity $\phi$ (see Supplementary Discussion, Section 2.1.2).

In Fig. 2a–c, we highlight the regions where either or both forms of social structure need to be included in the model to accurately capture each of the three outputs of model AH. Those regions are then combined in Fig. 2d into an overall simplest-model-acceptance-region plot, in which a less-structured model is rejected when at least one of the outputs is too inaccurate. Model U appears accurate for both the final size $z$ and peak incidence $\pi$ in a diagonal region of the parameter space, i.e., when the increase

owing to stronger within-household transmission and the decrease owing to larger age differences cancel each other out. The graph for the time to the peak $t$ is more complex, and highlights the presence of a region where at least one structure, no matter which, is needed, but not necessarily both (green area in Fig. 2c).

Four different parameter sets are then arbitrarily chosen, one in each region of the overall plot in Fig. 2d (orange labels e–h). For each choice, Fig. 2 (respectively, panels e–h) compares the full incidence and cumulative incidence curves of the models and suggests that the regions given in Fig. 2d indicate when models U, A and H approximate well not only the three outputs of interest but also the full dynamics of model AH.

**Sensitivity analysis.** Finally, we explore how results depend on the remaining two parameters of model AH. Figure 3a reports the overall simplest-model acceptance regions for varying values of $R_0$ and the relative infectivity of children versus adults $\phi$, still under baseline random mixing. For increasing $R_0$ (left to right), the household structure becomes less relevant for accurate predictions, unless within-household transmission is extremely high. Intuitively, at least for the final size, this is because at large $R_0$ almost all the population ends up infected, leaving very little room for stronger within-household transmission to increase it further (Fig. 1a). Despite not being capped, as the final size is by the total population, the peak incidence shows a similar behaviour. Figure 3a confirms the intuition that, under random mixing, the age structure can be neglected when age classes do not differ much in susceptibility and that the household structure can be neglected when not much transmission occurs in households. Quantifying this intuition, however, is an important result of this analysis. On the other hand, when realistic UK contact patterns[25] are assumed, as in Fig. 3b, the age structure becomes absolutely crucial for accurate predictions, whereas the household structure remains important only at large values of within-household transmission.

To facilitate intuition about the quantitative meaning of Fig. 3, we locate points in the parameter space corresponding to some scenarios inspired by real infections. Scenario 1 is characterised by parameters in the ballpark of H1N1 2009 pandemic influenza (points 1): $R_0 \approx 1.5$, $\phi \approx 1$, $\psi \approx 2$[33] and values of $p_{aa}$ from 0.1 to 0.4, corresponding to the wide range of SAR estimates found in the literature[35], from <10%[40] up to 40%[35]. At the low end of this range, Fig. 3 suggests the household structure can be neglected, whereas at the high end the household structure should be retained, at least when mixing is heterogeneous (Fig. 3b, points 1). Scenario 2 has parameter values closer to those estimated for the 1918 influenza pandemic[34], with a higher $R_0$ than in 2009, and children apparently not more susceptible but potentially more infectious than adults. We choose $R_0 \approx 2$, $\phi \approx 2$, $\psi \approx 1$ and $p_{aa} \approx 0.3$ (leading to a SAR of ~30%[34]) and conclude that the household structure can (just about) be ignored, whereas the age stratification is necessary when mixing is heterogeneous (Fig. 3b, point 2). However, we note that the higher transmissibility seen in the 1918 influenza pandemic compared to the 2009 one might be partially owing to a different demography and social structure, in particular with larger households. For Scenario 3, we imagine the hypothetical invasion in a fully susceptible population of a highly transmissible infection, with characteristics similar to common childhood infections like measles or chickenpox: $R_0 \approx 4$, $\phi \approx 1$ and a trade-off between $\psi$ and $p_{aa}$ owing to the fact that only child-to-child estimates of transmission probability are available in the literature (see Supplementary Methods, Section 1.6.5). This leads to a range of possible points 3 in Fig. 3, all of which suggest the household structure can be ignored, whereas the age stratification is essential. Applications to other topical emerging infections

(e.g., SARS, MERS, Ebola) would also be possible, though data scarcity and other limitations highlighted in the Discussion suggest conclusions in these cases would be rather tentative. Furthermore, parameter values for SARS and Ebola, with their marked lower susceptibility of children compared with adults[41], would not appear in the range displayed in Fig. 3.

**Rule of thumb**. Overlaying the contour lines for the SAR onto Fig. 3b reveals how the SAR can be used directly (i.e., without estimating $p_{aa}$) to discriminate whether the household structure is needed or not, somewhat irrespective of the relative infectivity and susceptibility of children, but strongly depending on the value of $R_0$. This dependence is explored more extensively in Fig. 4a, where the values of the SAR along the lines at which model A differ by 5% in relative terms from model AH (the borders between the light blue and red regions in Fig. 3b) are collected for all values of $\psi$ ranging from 1 to 4 and for $\phi = 1$, 1.5 and 2, and plotted against $R_0$. The relationship appears approximately linear (Fig. 4a) for a wide range of values of $R_0$, except for $R_0$ close to its threshold value of 1 or for a SAR roughly above 70%. Figures 3b and 4 together allow the formulation of an approximate but simple empirical rule of thumb:

For accurate predictions of expected epidemic final size, peak daily incidence and time to peak daily incidence in a developed country with UK-like contact patterns relevant for respiratory infections, the age structure should always be included in the model and the household structure should be present if SAR ≥ $(\eta_1 \times R_0 + \eta_0)$%, where $R_0$ can be estimated from the early growth with standard methods (i.e., in the absence of households), $\eta_1 = 24$ and $\eta_0 = -10$.

The coefficients of this linear relationship have been determined empirically to carry more weight from values of the SAR associated to more moderate and realistic differences in susceptibility and infectivity of children versus adults (see Fig. 4 and Supplementary Discussion, Section 2.4), and to provide simultaneously a satisfactory fit for both UK-like mixing patterns (Fig. 4a) and random mixing (Supplementary Fig. 25B). The term "accurate" refers to predictions with a theoretical relative accuracy of 5%, given perfect observations and parameter estimates, and assuming model AH is the truth. The possibility of using standard methods of estimating $R_0$, e.g., from the exponential growth rate $r$, comes from the observation[42] that ignoring age and household structure does not fundamentally undermine the accuracy of such estimates (see also Supplementary Methods, Sections 1.1.2 and 1.1.3).

Despite its simplicity, the linear relationship between the SAR and $R_0$ is surprisingly solid, given the amount of complexity captured. In particular, it appears not to be significantly affected by contact patterns (see Supplementary Fig. 25), and is therefore still informative about whether the household structure is needed or not even in the case of random mixing or other intermediate contact patterns, suggesting it can be broadly exported to other developed countries; and it extends to very different social structures (e.g., Sierra Leone, Fig. 4b) and to other accuracy thresholds (Supplementary Fig. 25), though the line coefficients require suitable modifications investigated in the Supplementary Discussion (Section 2.4).

Finally, although in the rule of thumb described above for a developed country with UK-like contact patterns the age stratification appears essential for almost all parameter values explored ($\phi = 1$, 1.5 and 2, $\psi$ from 1 to 4; $R_0$ from 1.1 to 4), this is not always the case in general, e.g., with random mixing, a 10% threshold or other social structures (see Supplementary Discussion, Sections 2.1.2, 2.1.3 and 2.3.4). Although a clear condition determining when the age structure is needed for accurate

predictions seems less evident to formulate, we deem it also less essential, given the limited complexity the age structure component adds to the model and the wide availability of age-stratified epidemiological data.

## Discussion

The main aim of this analysis is to provide a deeper understanding of the role played by structural model components on epidemiological predictions. For this reason, unlike multi-model ensemble approaches[18,23] where models' outputs are combined into a median or weighted average[43], here outputs are kept separate[44] to highlight disagreements between models (in the same spirit as structured decision-making[17]). Furthermore, rather than only focusing on predictions in a single scenario, models are compared on a wide region of the space of observables (i.e., $R_0$, SAR and the ratio of adults and children in the incidence), and sharp boundaries are identified between regions in this space where models behave similarly and regions where they differ.

Overall, our analysis suggests that the age structure seems to be a more rewarding ingredient to add to the model than the household structure, especially, if measured against the respective cost in terms of mathematical complexity. In other words, if one is not interested in the explicit presence of households (details of local transmission, targeted interventions, etc.), their complex and antagonistic effects—longer and more frequent local contacts, but also stronger saturation effects—may often cancel out to a good approximation. This is more likely to be true for large values of $R_0$. However, it is worth emphasising that ignoring households systematically underestimates epidemic severity, predicting a lower final size, lower peak incidence and a later peak time.

The best retrospective example of how this study could have guided model design is the case of the H1N1 2009 pandemic, because of the early availability of data also at the level of households. With initial parameter estimates[33,45] as presented above ($R_0 \approx 1.5$, $\phi \approx 1$ and $\psi \approx 2$, somehow capturing prior immunity in the elderly) and average SAR values of the order of 10–13%[33], this study would have strongly discouraged the use of complex households models in favour of purely age-stratified ones. In fact, model A would have predicted a final size of 34% (compared with the 35% of model AH), which is much lower than the 58% of the homogeneously mixing model. Current estimates of the final size from retrospective analysis in Italy are even lower (in the range 15–30%[12,46]), but such discrepancy is easily imputable to interventions and spontaneous social distancing, and possibly a different demography and more refined age structure of these studies compared with the simple two-class one presented here. Later studies reported higher estimates for the SAR[35], but even with SAR values ~40% (corresponding to $p_{aa} = 0.4$), on the high end of the range displayed in Fig. 3b, model A would predict a final size of 37%, still very close to the 40% of model AH (a relative difference just above 5%).

These results apply in particular to the case of children being more susceptible and/or infectious than adults, which is the main one discussed here. The situation is more complex in the opposite case, as sometimes we have found the mapping from model AH to model A to fail (see Supplementary Discussion, Section 2.3.3). The insight is that, when children are less susceptible or infectious than adults, sometimes the mixing imposed by the household structure results in an age-stratified incidence that cannot be reproduced by an age-stratified model alone (with the same contact rates), irrespective of the level of assortative mixing assumed. In this sense, the household structure becomes essential to reconcile observed disease dynamics and measured contact rates of adults and children. More research, in particular, on the implications for public health, is needed in this direction. Except

in this region of the parameter space, though, the household structure does appear in general less relevant for accurate predictions than the age structure (Fig. 3). Throughout, we have assumed a 5% accuracy threshold, which is probably optimistic in an emerging epidemic scenario. However, the same result appears even more strikingly at less-stringent accuracy requirements, as the need for household structure drops, whereas the age structure, at least with heterogeneous and assortative mixing patterns, remains essential (see Supplementary Discussion, Section 2.2.2 and Supplementary Fig. 16).

At a methodological level this work highlights how, even in a rather simple context, comparing models that are structurally different—i.e., differing in the number and biological meaning of parameters—is not a straightforward and unambiguous process, and requires making choices on which the conclusions may fundamentally depend, in particular, what to keep fixed across models (we argue these should be observable or directly estimable macro-parameters with a model-independent biological interpretation); how model-specific parameters are derived from these observables (a process that might unravel numerous unidentifiability issues); and how models' predictions are compared.

The choices we made here, though arguably natural, are not unique. For example, we decided to focus only on predictions in terms of the three selected outputs. Despite being chosen for their public health relevance, other outputs might be of more interest, depending on the question addressed. We then only presented results in terms of aggregate outputs (some comments on age-stratified final size appear in the Supplementary Discussion, Sections 2.1.2 and 2.2.2). Furthermore, we also exploited the fact that model AH includes all the others as submodels and we were therefore able to compare predictions by studying the divergence of the outputs of simpler models from the assumed truth of model AH. Alternative choices would be required in the absence of a full model. Finally, we chose to keep $R_0$ fixed across all model, whereas another perhaps even more natural choice would have been to fix the real-time growth rate $r$. This would have been feasible in the case of models involving constant transition rates between compartments (analytical methods for computing $r$ in the presence of household can be found, e.g., in[8]), but the lack of exact results in the case of the time-since-infection modelling framework assumed here (see Methods and Supplementary Methods, Sections 1.1.3, 1.2.9 and 1.4) would have caused significant loss in computational efficiency. A brief exploration (only in the baseline scenario: see Supplementary Discussion, Section 2.3.2) leads to similar results, with simpler models appearing even more likely to be accurate than the constant-$R_0$ comparison suggests. This is in line with previous results[42,47] showing that ignoring the household structure in the time-since-infection model used here leads to an overestimation of $R_0$: models with households, therefore, when forced to match the same $R_0$, need to compensate with a slightly larger overall infectivity, and this leads to a larger final size and peak incidence, if instead the models are calibrated on the same $r$, no compensation in infectivity is needed and models with no households are likely more accurate on a larger region of the parameter space (see Supplementary Discussion, Section 2.3.2). However, not all aspect of the present study can be captured by the simple comparison of $R_0$ and $r$[42,47], given past work is restricted to the exponentially growing phase, whereas the present study investigates also the late epidemic behaviour.

Numerous limitations can be highlighted. The first and most obvious one is the lack of uncertainty: despite models being stochastic, the comparison is performed by deterministically mapping parameters from one model to another, thus ignoring the impact of any randomness, noise or estimation bias on model observables and outputs; in addition, final predictions are deterministic. It would not be difficult to compare the full distributions of model outputs, but more complicated results than binary acceptance/rejection of simpler models based on arbitrary thresholds would likely be harder to present and interpret. A more significant step would be to implement uncertainty in the model-mapping process. However, the choice we made in this study was to trade proper uncertainty management for the ability to perform extensive exploration of the space of observables, as any formal statistical estimation procedure, especially, if Bayesian in nature, would have been computationally challenging, and possibly difficult to automatise. Nevertheless, the role of uncertainty could still be roughly quantified by heuristically considering confidence intervals around each quantity in Fig. 3 (or similar figures, for values that do not appear there), given that boundaries appear to be changing monotonically with parameters: for example, for confidence intervals (1.5, 2) for $R_0$ and (20%, 40%) for the SAR, from Fig. 3 one can still conclude the clear need for age, and the need for household structure in the lowest $R_0$ range only when SAR >30% and with decreasing and ultimately vanishing need as $R_0$ increases ($\phi$ and $\psi$ are almost irrelevant).

A second limitation is that we only distinguished between adults and children. Stratification to more than two age groups could lead to some non-trivial unidentifiability problems in mapping relative infectivities and susceptibilities from age-stratified incidence, and would require to rely more extensively on data about age-specific contact patterns[25], which has been here kept to a minimum to facilitate generalisability to different population structures. Furthermore, with more parameters, results would be again more difficult to present, so it was deemed less relevant in the first instance. However, for practical purposes, for example, in the case of influenza, it would be relevant to have at least a third age group, i.e., seniors, who make up a substantial fraction of developed countries' population and typically have lower contact rates[25], potentially reduced susceptibility owing to prior immunity[33,36,48] and higher risk of severe symptoms[49].

Third, we assumed a fully susceptible population, thus fundamentally restricting this framework to emerging infections. Changes in mixing and transmission parameters could account for the limiting cases of perfect (i.e., fully protective) prior immunity distributed uniformly in the population or affecting entire age classes, but a more general and realistic case would require to know how prior immunity is distributed in households. Alternatively, a mechanism to generate such prior immunity (e.g., multiannual simulations run prior to the start of the epidemic) would be needed, but then results would depend on the details of such a complex approach.

Fourth, the assumption of constant parameter values, though essential for analytical results, effectively makes model predictions approximately valid only for fast and relatively mild infections, influenza being the most natural example. Paradoxically, though, most new infections for which the emerging epidemic scenario considered here would be relevant are often highly pathogenic (e.g., SARS[41], MERS[50], Ebola[51]), thus eliciting active interventions or spontaneous behavioural change. Outbreak response would likely reduce contact rates over time, (as well as time intervals to notification or hospitalisation[52]), causing the actual epidemic to deviate from initial forecasts. Analytical results are unlikely to be available for non-constant parameters, but we note that inclusion of assumed changes in parameters over time owing to explored interventions would not compromise the computational efficiency of this approach (they would affect only forward predictions, not the mapping procedure itself—see Methods).

Fundamentally, though, predictions in the presence of interventions are not the main focus of this analysis for two reasons:

first, the choice of which interventions to explore would strongly shape the model selection process; and second, the models considered here are arguably too simple for most realistic interventions. To study interventions in specific scenarios, more tailored models and more statistically principled methods for robust uncertainty quantification should be used. structures on epidemic predictions in the absence of interventions, and should be viewed as complementary to these other methods. However, such early predictions would still be highly informative in assessing the impact of interventions (or additional impact after estimates became available) by comparison with the epidemic outcome actually observed. Finally, note that potential factors playing a role only under certain intervention scenarios would not be useful to tease apart models compared in the absence of such interventions; however, this is not a limitation of the current methodology, but rather a typical out-of-sample prediction problem[17].

Fifth, the assumption of random mixing within the household could also be questionable in certain contexts. The data used for age-specific mixing patterns[25] suggest a focus on respiratory infections, for which within-household random mixing appears reasonable[53], but applications to other directly transmissible infections might require different assumptions. Extensions to within-household age-stratified mixing would be possible, but would require estimates dependent on the household composition (see Supplementary Methods, Section 1.2.3), or potentially even disease-specific (e.g., school-age children less at risk to Ebola[54], possibly because less likely to care for sick individuals[55]) and hence likely unavailable and hard to collect prospectively. But even age-stratified mixing would fall short of other potentially important heterogeneities in within-household contact patterns, e.g., specific family members taking on the responsibility of caring for relatives infected with Ebola[55].

Sixth, we have focussed on age and households only. Other forms of social structure exist, which in some cases could be more relevant and even reasonably well informed by data (e.g., hospital[56] or funeral transmission[57]). In addition, we have ignored any network, patch or explicit spatial structure, or other even more complex models, owing to scarcity in data, analytical results or both.

Finally, we have assumed the presence of a clearly defined exponentially growing phase. This was not the case for MERS and SARS, owing to small outbreak sizes and/or strongly spatial and localised (hospital-based) spread[56], thus questioning the reliability of estimates of $R_0$ and other parameters used here, and hence compromising the applicability of our mapping procedure.

Despite all limitations, though, in the restricted context of an emerging infection that is fast enough or not too severe (e.g., influenza—or Zika, though social mixing patterns are likely less relevant for vector-borne diseases) our comparison can inform a useful trade-off between model simplicity and prediction accuracy. Improvements are certainly possible, but we believe the general approach proposed here should be adopted for comparing models and shed light on how structural model components affect predictions. In the particular case of age and household structure, substantial information is still needed early on in an outbreak of a new emerging infection for an appropriate, scientifically driven model choice. Nevertheless, the present analysis suggests that, in many real-world scenarios, households give limited additional accuracy in overall descriptors of epidemic severity for high cost in terms of model complexity and, despite the risk of slightly underestimating the epidemic severity, are probably worth neglecting in the first instance, when limited information is available and rapid predictions are needed.

## Methods

**Aspects common to all models.** All models share the same stochastic, time-since-infection transmission process[8,58]: each infected individual makes infectious contacts at a rate described by a function $\beta\omega(\tau)$, where the total infectivity $\beta$ depends on the ages of the infector and the other contacted person, as well as on the environment (i.e., within or outside the household), and the infectious contact interval distribution $\omega(\tau)$[59] is a function of the time since infection $\tau$, normalised to integrate to 1, with mean $T_G$ (often referred to as the generation time[37]) and is the same between every pair of individuals, irrespective of age and environment (see Supplementary Methods, Section 1.1.2). After infection, individuals cannot be infected again.

The dynamics of this model, at least in its deterministic form and in the absence of household structure, could be described in terms of a renewal-type integral equation[8,58], with $\beta = R_0$, but we do not present it here because no dynamical equations are solved in this work. Instead, for all models, the observables defined during the exponentially growing phase and used for the mapping (i.e., $R_0$, the incidence ratio of adults and children and the SAR), as well as the final size $z$, are computed directly using available mathematical results, whereas the peak incidence $\pi$ and time to the peak $t$ are computed as the average of 100 individual-based stochastic simulations in a population of 100,000 individuals. The epidemics are started with $n_0 = 50$ initial cases, to avoid stochastic extinction and minimise the effect of random delays at the start of the epidemic, and are synchronised at the peak.

Inspired by influenza, we choose a $\Gamma$-shaped infectious contact interval distribution $\omega(\tau)$ with mean $T_G = 2.85$ days and shape parameter $\alpha = 9$[8,32]. However, this particular choice does not affect the quantities calculated analytically (observables and final size), which are all time-integrated quantities, and hence bears no influence on the mapping procedure, although it does affect the peak incidence $\pi$ and the time to the peak $t$. However, to allow generalisations to other infections, the time to the peak is rescaled by a factor $T_G$, thus approximately measuring the average number of generations to the peak.

Although the stochastic simulations are computationally intensive, the forward problem of computing observables and outputs for each model is relatively inexpensive even in the absence of analytical results, as it needs to be performed once for each combination of basic model parameters. Instead, the inverse problem of exploring parameter spaces at fixed observables is in general computationally expensive. In the presence of explicit expressions for the observables, which is rarely the case, one could simply invert them to map parameters directly from one model to another[42]. In the absence of explicit expressions, iterative methods must be used (see Supplementary Methods, Section 1.1.1), which require solving the forward problem multiple times and might become prohibitive for computationally costly simulations. The efficiency of our approach stems from the fact that the context we have focussed on, though rather specific, is rich in analytical results: in particular, thanks to the latest methodological advances in the computation of $R_0$ for households models[47,60], the observables of all models can be obtained without having to integrate the system dynamics. Therefore, the mapping procedure could be performed on all points of a regular grid in the parameter space of model AH (Figs. 1, 2a and 3) that is fine enough to suggest our conclusions, though numerical in nature, hold throughout the explored portion of space.

**Model AH (age and households).** Model AH is parameterised as follows. The infection spread between adults ($a$) and children ($c$) in each environment $x$ (in the community, $x = g$ for "global"; or in the household, $x = h$) is parameterised in terms of a next-generation matrix (NGM) of the form

$$K_x = \begin{pmatrix} k_{aa}^x & k_{ac}^x \\ k_{ca}^x & k_{cc}^x \end{pmatrix} = \beta_x \begin{pmatrix} \gamma_x - \frac{N_c^x}{N_a^x} & (1-\theta_x)\phi \\ \psi(1-\theta_x)\frac{N_c^x}{N_a^x} & \psi\theta_x\phi \end{pmatrix} \quad (1)$$

(derivation in Supplementary Methods, Sections 1.1.4 and 1.1.5), where $k_{ij}^x$ gives the average number of infectious contacts an individual in age-class $j$ makes with individuals in age-class $i$ in environment $x$. In the initial phase of the epidemic all the infectious contacts $k_{ij}^g$ in the community lead to real infections. In a household, instead, some of the $k_{ij}^h$ infectious contacts hit previously infected or immune cases.

The NGM $K_x$ incorporates simultaneously both contact and transmission elements. The contact patterns are given by: $\gamma_x$, the ratio of the numbers of daily contacts an adult and a child have in environment $x$; $N_a^x$ and $N_c^x$, the numbers of adults and children in environment $x$, respectively; and $\theta_x$, the assortativity of children in $x$, defined somewhat non-standardly as the fraction of contacts that a child makes with other children in $x$ and ranging from 0 (fully antiassortative mixing) to 1 (fully assortative mixing). Random mixing is achieved for $\theta_x$ equal to the fraction of other children in the environment. Within-household mixing is always assumed to be random (note that this requires $\theta_h$ to depend on the household composition). We assume frequency-dependent contact patterns in the households, so that the infectious contacts $k_{ij}^h$ are distributed among all (other) cases of age-class $i$; that is, in the simple case of all identical individuals, the person-to-person contact rate in a household of size $n$ scales as $1/(n-1)$—see Supplementary Methods, Section 1.2.3, for precise age-stratified details.

The transmission component of the model is parameterised in terms of $\psi$ and $\phi$, which represent respectively the relative susceptibility and infectivity of children versus adults, and total infectivities $\beta_x$. In practice, the within-household total infectivity $\beta_h$ is re-parameterised in terms of $p_{aa}$, defined as the probability that a randomly selected susceptible would be infected directly by a single initial household case, in a randomly selected household with at least two individuals, when adults and children have the same susceptibility and infectivity ($\psi = \phi = 1$). In other words, $p_{aa}$ is obtained by first computing $p_n = 1 - \exp(-\beta_h/(n-1))$ and then averaging $p_n$ over the size distribution of a randomly selected household with at least two members. Other similar choices would have been possible, as long as they do not depend on other parameters we are exploring independently, like $\psi$ or $\phi$. In the Supplementary Discussion (Sections 2.1.1 and 2.2.1) we comment on more aggregate, but more intuitive epidemiological quantities, such as the SAR or the fraction of total transmission occurring in households. The latter is measured as $(R_0 - R_0^g)/R_0$, where $R_0^g$ is the dominant eigenvalue of $K_g$, and reveals that, at least for the H1N1 2009 pandemic influenza in Great Britain, approximately a third of the total transmission occurs in household (a rule of thumb suggested before[8,9,32]; see Supplementary Discussion, Sections 2.1.1 and 2.2.1).

Numerical values are as follows. At baseline, the population structure is that of Great Britain[61], with a fraction $F_c = 22.73\%$ of the population consisting of children and a mean household size $\chi = 2.35$ (see Supplementary Methods, Section 1.6.1, and Supplementary Tables 1–4). Other social structures (South Africa: $F_c = 45.92\%$, $\chi = 4.27$; Sierra Leone: $F_c = 53.81\%$, $\chi = 5.85$) are explored in the Supplementary Methods, Section 1.6.2, Supplementary Tables 4–7 and Supplementary Discussion, Section 2.3.4. At baseline, contact patterns assume random mixing: $\gamma_h = \gamma_g = 1$ (adults and children have the same contact rates everywhere) and $\theta_g = F_c = 22.73\%$, the fraction of children in the population. Parameters for UK-like contact patterns, characterised by strongly assortative mixing, are estimated from the POLYMOD study[25] to be $\gamma_h = \gamma_g = 0.75$ and $\theta_g = 58\%$ (see Supplementary Methods, Section 1.6.3, and Supplementary Table 8), and are used also for other social structures (in the absence of contact pattern data for South Africa and Sierra Leone). Intermediate contact patterns are explored in the Supplementary Discussion (Section 2.3.1).

The observables for model AH are derived as follows. The basic reproduction number $R_0$ is computed using a multitype extension of the technique developed in[60] that leads to the construction of a suitable matrix $M$ (details in the Supplementary Methods, Section 1.2.4), the dominant eigenvalue of which is $R_0$.

From $M$ it is also possible to reconstruct the vector $v^{AH} = (v_a, v_c)^\top$ (superscript $^\top$ denotes transposition, to give a column vector), whose components are the fractions of adults and children in each generation ($v^{AH}$ is constant during the exponentially growing phase), correctly computed by taking into account both household and global transmission (Supplementary Methods, Section 1.2.5). Primary cases in households are infected globally, so arise in proportions given by the components of the vector $v_h^{AH} = (v_h^a, v_h^c)^\top$, obtained by renormalizing $K_g v^{AH}$ so that its components sum to 1.

The SAR is defined as the fraction of initial susceptibles that are infected in a within-household outbreak started by a single individual in a typical household infected during the exponentially growing phase. Its computation is not trivial, because the distribution of infected households during the exponentially growing phase is affected by the age-dependent between-household transmission: if children are more likely to be infected in the population, larger households are also more likely to be infected because they tend to contain more children. We denote by $\{\pi_n^a\}$ and $\{\pi_n^c\}$ the size distributions of the household of a randomly selected adult and child, respectively, and by $\mu^a$ and $\mu^c$ the average epidemic sizes in the household of a randomly chosen initial adult or child case, respectively. Then the average size of a household epidemic during the exponentially growing phase is $\mu^{AH} = v_h^a \mu^a + v_h^c \mu^c$. The household SAR is then computed as $(\mu^{AH} - 1)/(\chi^\gamma - 1)$, where $\chi^\gamma$ is the average size of a household infected in that phase (see Supplementary Methods, Section 1.2.6).

Finally, the outputs for model AH are derived as follows. The average final size $z$, in the asymptotic limit of an infinite number of households, is computed using the methodology described in ref. [30] (Supplementary Methods, Section 1.2.7). The peak incidence and the time to the peak are obtained from individual-based stochastic simulations in a synthetic population with the required social structure and contact patterns (Supplementary Methods, Sections 1.2.8 and 1.5). To minimise the convergence time from the initial conditions to the stable proportions of cases of each type during the exponentially growing phase, the $n_0$ initial cases are all chosen as primary cases in different households and consist of adults and children in proportions given by $v_h^{AH}$.

Further details about model AH can be found in the Supplementary Methods, Section 1.2.

**Model A (age).** Model A is parameterised as follows. The spread between adults and children in model A is described by $K^A$, a NGM of the same form as the one in Eq. (1), but with elements indexed by A (there is only one environment). The contact patterns are given by $\gamma^A$, the ratio of the overall number of contacts an adult and a child have, and the assortativity of children $\theta^A$. The transmission component of the model is parameterised in terms of an overall transmission parameter $\beta^A$ and the relative susceptibility and infectivity of children versus adults, $\psi$ and $\phi$, which are thought of as biological parameters ideally accurately measured via detailed household studies, and are therefore assumed to be the same as in model AH. The presence of four parameters, two of which coincide by construction with those of model AH, makes model A both more flexible and more tightly linked to model AH than to model H (two parameters only). This partly explains why model A is better than model H at mirroring the outputs of model AH in Fig. 1.

Numerical values are inherited by the social structure and mixing patterns of model AH. At baseline (Great Britain[61]) the fraction of children is 22.73% and adults and children have the same contact rate ($\gamma^A = 1$). UK-like contact patterns are given by $\gamma^A = 0.75$. The assortativity $\theta^A$ is estimated in the mapping procedure (see below).

In terms of observables, the basic reproduction number $R_0$ in model A is computed as the dominant eigenvalue of the NGM $K^A$, and the corresponding eigenvector $v^A$, normalised to have components summing to 1, represents the fraction of adults and children in each generation[58].

In terms of outputs, the final size is computed using standard methods[5] (Supplementary Methods, Section 1.3). The peak incidence and the time to the peak are again computed using the individual-based stochastic simulation, with no household structure and starting with $n_0$ initially cases, consisting of adults and children in proportions given by the components of the vector $v^A$ (to start as close as possible to the stable proportions of adults and children during the exponentially growing phase).

Further details about model A can be found in the Supplementary Methods, Section 1.3.

**Model H (households).** The pure households model is parameterised in terms of a global total infectivity $\beta_g^H$ and a within-household total infectivity $\beta_h^H$, representing, respectively, the average number of infectious contacts an infective makes in the community and in their household, during their entire infection period. Early on in the epidemic, every infectious contact in the community leads to a new infection. Frequency-dependent contact patterns are assumed within the household, so that the number of infectious contacts toward a single member of a household of size $n$ is $\beta_h^H/(n-1)$.

Numerical values are again inherited by the household structure of model AH. At baseline, the household size distribution is that of Great Britain, with a mean household size $\chi = 2.35$ (see Supplementary Tables 1–3). Other social structures are also considered (South Africa: $\chi = 4.27$; Sierra Leone: $\chi = 5.85$).

The computation of $R_0$ for model H follows the method of[47,60] (Supplementary Methods, Section 1.4). Similarly to model AH, the SAR is computed as $(\mu^H - 1)/(\chi - 1)$, where $\mu^H$ is the average size of a within-household epidemic, computed using standard methods for small populations[5,62] and $\chi$ is the average household size. However, care needs to be taken in the choice of the correct household size distribution (see below).

The final size is computed using standard analytical techniques[28], and the peak incidence and time to the peak are obtained from stochastic simulations starting with $n_0$ cases, all primary cases in different households, divided in adults and children according to the components of $v_h^{AH}$ (to start as close as possible to the stable household size distribution of the exponentially growing phase).

Further details about model H can be found in the Supplementary Methods, Section 1.4.

**Model U (unstructured).** Given the temporal details of the infection process are fixed by the infectious contact interval distribution $\omega(\tau)$, the model with pure homogeneous mixing has only one parameter $\beta = R_0$. The final size is computed standardly as the only positive solution $z$ of $1 - z = e^{-R_0 z}$ [5], whereas the peak incidence and the time to the peak are obtained via simulations starting with $n_0$ initial cases.

**Model-mapping procedure.** For each combination of basic parameters for the assumed-true model AH ($\psi$, $\phi$, $p_{aa}$, from which $\beta_h$ is derived, and $\beta_g$—as well as fixed $\theta_g$, $\gamma_g$, and $\gamma_h$—we calculate the true epidemic observables $R_0$, $v^{AH}$, and SAR, as described above. In practice, the parameter space in all figures is explored at constant $R_0$, so that for each choice of $p_{aa}$, $\psi$, and $\phi$ we compute the value of $\beta_g$ required to achieve a desired $R_0$. These observables are then used to map the parameters for the other models as follows.

We start by mapping model AH to model A. Parameters $\psi$ and $\phi$ in model A are assumed to be known and the same as in model AH. Then $\theta^A$ is ideally chosen to match $v^{AH}$. Unfortunately, there are parameter values for which no suitable value of $\theta^A \in [0, 1]$ can be found (see Supplementary Methods, Section 1.3). This is often the case for $\psi < 1$ (Supplementary Discussion, Section 2.3.3). The overall infectivity $\beta^A$ is then chosen to match $R_0$. There are no households in model A, so the SAR is not used.

To map model AH to model H, first $\beta_h^H$ is computed to match the observed household SAR. The correct household size distribution to use cannot be computed from model H alone, because the distribution of infected households during the exponentially growing phase is affected by the age-dependent transmission as described for model AH. In real scenarios, the within-household infectivity is measured from household studies. In such surveys, the recruitment of households

is subject to many constraints, but it ideally monitors a representative portion of the population of infected households. If model AH were an exact description of reality, then households would be recruited with size distribution $\{\pi_n^v\}$, where, for each $n$, $\pi_n^v = v_n^a \pi_n^a + v_h^c \pi_n^c$. In practice, instead of matching the same SAR as in model AH, we equivalently compute $\beta_h^H$ by imposing $\mu^H = \mu^{AH}$, with household size distribution $\{\pi_n^v\}$ (and hence $\chi = \chi^v$). The global infectivity $\beta_g^H$ is then computed to match $R_0$. Apart from appearing in the computation of the correct household size distribution for matching the SAR (rather than obtaining such a distribution from a random sample of infected households), $v^{AH}$ is not explicitly used.

Finally, the mapping from model AH to model U is trivial, as model U is only parameterised in terms of $R_0$ and the other observables are not used.

**Reporting summary**. Further information on research design is available in the Nature Research Reporting Summary linked to this article.

## Data availability

Household data for Great Britain (England, Wales and Scotland) are available from 2001 UK census data, table number C0844[61]; source: Office for National Statistics licensed under the Open Government Licence v.1.0. The table can be downloaded from https://doi.org/10.5281/zenodo.3629873 or https://github.com/lorenzo-pellis/model-mapping, or can be requested directly from the Office for National Statistics. UK social contact data are obtained from the POLYMOD study[25] (raw data available from the authors upon request). Household data for Sierra Leone are available from the 2008 Sierra Leone Demographic and Health Survey[63] and household data for South Africa from the 1998 South Africa Demographic and Health Survey[64]. Both can be requested from the Demographic and Health Surveys Program[65] (free registration). Raw data are only used to generate Supplementary Tables 1, 4, 6 and 8 (see Supplementary Information), from which all other tables, analyses and figures are generated.

## Code availability

Numerical codes, output files and figures have been archived at the time of publication with https://doi.org/10.5281/zenodo.3629873. Potential improvements or corrections will be made available at https://github.com/lorenzo-pellis/model-mapping. The main code to perform the model mapping is written in MATLAB and has been tested on both R2016a on Mac OSX El Capitan and R2019a on Microsoft Windows 7 Professional (Service Pack 1). The individual-based stochastic simulation is written in C++ and source codes are freely available.

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

## Acknowledgements

We acknowledge the MRC Methodology Grant G0800596. L.P. acknowledges the EPSRC (grant EP/J002437/1) and the Wellcome Trust and Royal Society (grant 202562/Z/16/Z); S.C. acknowledges funding from the NIGMS MIDAS initiative, the French Government's Investissement d'Avenir program, Laboratoire d'Excellence "Integrative Biology of Emerging Infectious Diseases" (grant number ANR-10-LABX-62-IBEID), the AXA Research Fund and Grand Prix Robert Debré; and N.M.F. acknowledges Centre funding from the MRC and DFID, Health Protection Research Unit funding from NIHR, Institute funding from Community Jameel and grant funding from the Bill and Melinda Gates Foundation. We also thank Niel Hens for support with the analysis of the POLYMOD data, the members of the Zeeman Institute at the University of Warwick for useful discussions, and the anonymous reviewers for their insightful and constructive comments.

## Author contributions

L.P., S.C., N.M.F., C.F. designed the study. L.P. developed the mathematical methods and numerical codes, performed the analyses and wrote the paper and supplementary material. S.C., N.M.F. and C.F. commented on the manuscript.

## Competing interests
The authors declare no competing interests.
