## [Peer Review File · Nature Communications]

Reviewers' comments:

Reviewer #1 (Remarks to the Author):

I found this to be an interesting analysis, however, I was struck by some significant limitations.

1. On Line 54, the authors state that they assume that some estimates are available from early data of key quantities, notably R_0 . If I am reading correctly, these are then used to populate the models. Though, in practice, these would be estimates (with associated uncertainty), and those estimates would likely improve in quality as a given outbreak is observed for longer. It appears that the authors are assuming that these early estimates are available and correct from the start, which seems quite limiting to any real world setting. There's an inherent catch-22 in the estimation of parameters from the outbreak itself, because those estimates will improve over time -- but of course, as time passes, the utility of projections declines since the outbreak itself has now been revealed.

I appreciate the general message of the authors, that there may be small marginal benefits to increasing model complexity, especially if doing so comes at a significant research cost. However, it is unclear to me why the method presented here, which seeks to highlight a best, or better, model structure is inherently better than a more standard ensemble approach (as mentioned in Lines 27-37). While it is true that the "wrong" model might fit early data well, the authors' presumption that such an approach would choose the "simplest model" is predicated on the assumption that the goal is to choose a single model. Rather, ensemble approaches, which have been used to great effect in other disciplines seem to overcome this problem.

2. The authors advocate the use of models here strictly for prediction (e.g. of epidemic size, peak timing, etc) and ignore the use of models for assessing interventions. This is relevant, as early predictions frequently are not borne out exactly because of interventions that are implemented. This was highlighted in the criticism of models used early in the 2014 Ebola outbreak, the early predictions of which were not borne out because they did not account for the significant efforts to respond to the outbreak. Indeed, simple prediction can still be very useful in generating support for interventions -- however, it would seem that this analysis would be more generally applicable if the authors were to think beyond the initial predictions and evaluate the impact of model structure on the prediction of the impact of interventions -- in the latter case, this could be considered in terms of the absolute consequences of an intervention (e.g. is it cost effective/) or the relative consequences of alternatives (e.g. do I need a more complicated model to pick the best intervention?)

3) Though the authors invoke Ebola, SARS, Zika and other recent high-profile outbreaks they then, in the discussion (beginning on L350) they then highlight that this approach is likely not relevant to these for specific reasons. It would be helpful to see a clearly motivated use case where this approach would be helpful; e.g. for what kind of decision-making would this solve a critically missing piece of the puzzle. And, per comment 1 above, would this approach be sufficient, or would one still want some other multi-model, or ensemble, approach to address the subsequent question about how to use model predictions to inform choices among alternative interventions. Again, per the authors' comments on L54, it is entirely possible that the model structure chosen here as best for an real outbreak would still not be sufficiently representative of the dynamics to accurately predict the consequences of interventions (e.g. because it misses a key dynamic that is not of great relevance in the absence of interventions). In such a case, one would then have done an analysis that conditions all future simulation on a single model that was only useful in the no-intervention setting.

In the end, I find that this is a very thorough analysis to illustrate a general approach to solving a model selection problem. My concern, however, is that it will be easy for some to interpret this as a final answer (e.g. use models with age, but not household, structure) rather than a protocol for

how to answer the question about which type of model to use.

Reviewer #2 (Remarks to the Author):

The current paper is concerned with model selection when analysing emerging epidemic outbreaks, and in particular studying what are the effects on prediction of choosing a simplified model.

This is an important topic and the paper contains some valuable conclusions (age-structure is more important than added within household transmission) based on an extensive simulation study. It is a welcome addition of an area not enough studied. I miss one paper which addresses the same type of question, but mainly from an analytical point of view: Trapman et al 2016 (Interface). I would have liked to see connections to conclusions there to see if they are in agreement or not (of course calibration is somewhat different).

I think you can explain a bit better on p2-4 how the simulation study is done. You simulate according to AH model which results in R_0 , incidence ratio and SAR. Then, depending on the model used in statistical analysis you use 1-3 of these quantities to calibrate the model, and then compute final size, incidence peak and time of peak. At least this is how I interpreted it ...

All results are based on numerics/simulations. Of course it seems likely to be valid conclusions in relevant parameter regions, but still analytical results would of course strengthen this. They are hard to obtain, but perhaps worth mentioning in discussion.

Minor comments

It is not clear to me why you study growth in terms of generations rather than in real time - the latter seems easier.

p2-3: R_0 for household models is complicated (refer to discussion later in manuscript)

Fig 1: The top row is the "truth"? If so this should be mentioned. Also, why no row for U model?

p7, 159: "though by not much"?

p8, 227: systematically underestimates ...: I think this is in agreement with Trapman et al.

p8, 229: then -> than

p9, middle: explain better what has been done mathematically

p10, 296: that that

p10: 331-332: "as long as ... not affecting exponentially growing phase". Why would it not?

p11: another thing not taken into account is spatial structure. This is of course harder, but still worth mentioning?

p11 bottom-12 top: first beta is expressed as a matrix (depending on age of infector + more) and then $R_0 = \beta$?

p14+p15: mention that when analysing using Model A SAR is not used, and incidence ratio not used when using model H

Response to reviewers: “**Systematic model selection for emerging epidemic predictions: the relative importance of age and household structure**”, by *L Pellis, S Cauchemez, N M Ferguson and C Fraser*

First, we would like to thank the reviewers for their insightful comments. They helped us being more precise both on the technical level and in how this work relates to other studies and existing methodologies. Their requests were pulling in different directions, no doubt reflecting two different examples of potentially interested audience, so we felt this reviewing process strengthened the manuscript in complementary ways. However, it also means it was difficult to address the comments of both Reviewers without adding a significant number of words.

Reviewer #1 (Remarks to the Author):

I found this to be an interesting analysis, however, I was struck by some significant limitations.

1. On Line 54, the authors state that they assume that some estimates are available from early data of key quantities, notably R_0 . If I am reading correctly, these are then used to populate the models. Though, in practice, these would be estimates (with associated uncertainty), and those estimates would likely improve in quality as a given outbreak is observed for longer. It appears that the authors are assuming that these early estimates are available and correct from the start, which seems quite limiting to any real world setting. There's an inherent catch-22 in the estimation of parameters from the outbreak itself, because those estimates will improve over time -- but of course, as time passes, the utility of projections declines since the outbreak itself has now been revealed.

The reviewer is correct: we have assumed point estimates are available and precise from early on in the epidemic. We recognise this is a simplification, as we agree with the reviewer that in real world settings estimates would (hopefully) come with associated uncertainty, and both estimates and uncertainty would change over time.

However, the reason for our choice is that accounting for uncertainty in the traditional sense was not the main aim of this analysis. Here, we have first of all tried to distinguish between basic model parameters and “observables”: the former are model-dependent parameters, which are often not observed directly; the latter, instead, are macro-parameters with a clear biological meaning that transcends the model-specific details, that would ideally be observed or easily estimated from data, and from which the basic model parameters would be estimated. The observables we work with are R_0 , the household secondary attack rate and the ratio of incidence in children vs that in adults (but we also briefly discuss in the SI the real-time growth rate), and they are thought of as “data”, i.e. directly observed. Therefore, in a real scenario of epidemic response, one would have only one specific value for the observables (ignoring data conflicts), and all models would be calibrated to match the same observables. This would result in estimates, with associated uncertainty, of the basic parameters for each model, which would then lead to forward predictions, also with uncertainty (due to stochastic dynamics *and* basic model parameter uncertainty). The approach we took in this work is to compare the behaviour of different models when fitted to a large range of potential observables, to highlight regions of the space of observables where models behave similarly and regions where they differ. The large number of model fitting procedures (fitting models A, H and U for each choice of parameters of the true model AH) is computationally costly, so we consciously traded off proper uncertainty management (in the basic model parameters and predictions) for the ability to perform an extensive exploration of the space of observables. We now clarify this concept in lines 49-55, 249-252 and 364-368.

However, our approach does allow some insight into the impact on uncertainty. An important outcome of our analysis is the apparent monotonic relationship between the boundaries separating regions where models behave similarly or differ and some parameters (such as R_0 and the relative susceptibility of children vs adults) and the approximate independence on other parameters (the SAR and the relative infectivity of children VS adults). Because we compare the models at the points of a regular grid over a large region of the parameter space, this monotonic relationship

allows understanding how intervals are mapped into other intervals, and hence, at least approximately, how robust the conclusions are to uncertainty in the estimates of epidemiological quantities observed. This is what we exemplify in lines 369-381.

I appreciate the general message of the authors, that there may be small marginal benefits to increasing model complexity, especially if doing so comes at a significant research cost. However, it is unclear to me why the method presented here, which seeks to highlight a best, or better, model structure is inherently better than a more standard ensemble approach (as mentioned in Lines 27-37). While it is true that the "wrong" model might fit early data well, the authors' presumption that such an approach would choose the "simplest model" is predicated on the assumption that the goal is to choose a single model. Rather, ensemble approaches, which have been used to great effect in other disciplines seem to overcome this problem.

Again, we do not disagree with the reviewer, and we have added numerous remarks about this point in lines 49-55, 239-252 and 414-424. We see the present analysis as complementary, rather than alternative, to the application of ensemble approaches, in the sense that we do not undermine the relevance of ensemble predictions for public health decision making, but we believe that the exploration of the space of observables mentioned above produces important insight on the role of different structural model components on epidemic outcome that could not be easily gained if only one value for the observables was used (e.g. one specific epidemic, or infection) and models' predictions were combined as done in standard ensemble approaches.

We also note that our analysis does not necessarily need to be read as suggesting the use of a "best" model: in fact, we do generate predictions from all 4 models. However, unlike what is customary with ensemble predictions, we do not merge the predictions of these models into a median or weighted mean prediction, but we rather keep them separate and compare them. When they are similar, we simply suggest the simpler model is accurate enough (the more complex model can still be used, but the results would be roughly the same). When they differ, we highlight this fact, rather than taking a mean/median of these two separate predictions. The advantage is a clearer understanding of model differences; the disadvantage is a potentially higher dimensional output than just a single prediction (and associated uncertainty). We then collapse this higher dimensional output into a single model choice for ease of presentation, but only on the ground of parsimony and the assumption that model AH is the truth. We clarify all this in lines 242-452.

2. The authors advocate the use of models here strictly for prediction (e.g. of epidemic size, peak timing, etc) and ignore the use of models for assessing interventions. This is relevant, as early predictions frequently are not borne out exactly because of interventions that are implemented. This was highlighted in the criticism of models used early in the 2014 Ebola outbreak, the early predictions of which were not borne out because they did not account for the significant efforts to respond to the outbreak. Indeed, simple prediction can still be very useful in generating support for interventions -- however, it would seem that this analysis would be more generally applicable if the authors were to think beyond the initial predictions and evaluate the impact of model structure on the prediction of the impact of interventions -- in the latter case, this could be considered in terms of the absolute consequences of an intervention (e.g. is it cost effective/) or the relative consequences of alternatives (e.g. do I need a more complicated model to pick the best intervention?)

The Reviewer raises a very interesting point here. However, although we have now added a section discussing interventions (lines 401-4013), we still maintain that this work should remain focussed primarily on predictions in the absence of interventions. The study of interventions would require more sophisticated models than those considered, and analytical results would be restricted to only limiting, and unrealistic, scenarios (e.g. full vaccination of one age group only). Furthermore, the choice of potential interventions would in itself guide the model structure, thus limiting the relevance of the model comparison study itself. Finally, if extensive exploration of the space of observables is already challenging, model comparison for any observables *and* any type and intensity of intervention would be close to unmanageable. We have therefore decided not to focus on interventions. However, as the reviewer points out, interventions (whether public health-

driven or due to spontaneous behavioural change) always affect the epidemic dynamics. Nevertheless, predictions in the absence of interventions are still essential to allow estimation of their effect by comparison with observed dynamics.

3) Though the authors invoke Ebola, SARS, Zika and other recent high-profile outbreaks they then, in the discussion (beginning on L350) they then highlight that this approach is likely not relevant to these for specific reasons. It would be helpful to see a clearly motivated use case where this approach would be helpful; e.g. for what kind of decision-making would this solve a critically missing piece of the puzzle.

The most natural example where prediction from this work would be directly useful is that of pandemic influenza, because the combination of fast spread and relative low severity limits the impact of control policies and spontaneous behavioural change. This example is now discussed in lines 262-279. The case of Zika could also be used, though much of the emphasis of the paper was on mixing patterns relevant for directly transmissible airborne infections, which would become irrelevant for Zika. For this, we left the current wording in place.

However, we have expanded the discussion to highlight the importance of predictions in the absence of intervention as a way to estimate the effect of interventions. With this perspective in mind, this work would be relevant for Ebola too. However, data about within-household transmission is limited, and we have also highlight how some aspects important for Ebola transmission (funerals; higher chances of transmission to family members taking care of infectives, and hence reduced risk of infection to children; bed capacity of treatment centres) have been neglected, thus making results from the specific models we have used here not particularly reliable in this case (lines 186-188 and 435-445). The main limitation in the cases of SARS and MERS is the lack of widespread community transmission (lines 186-188 and 451-455).

And, per comment 1 above, would this approach be sufficient, or would one still want some other multi-model, or ensemble, approach to address the subsequent question about how to use model predictions to inform choices among alternative interventions.

We now clarify in multiple places (lines 49-55, 239-252 and 414-424) why our approach differs from multi-model ensembles, that we see our study as complementary to such ensemble approaches, and that interventions are not the main focus of this work.

Again, per the authors' comments on L54, it is entirely possible that the model structure chosen here as best for an real outbreak would still not be sufficiently representative of the dynamics to accurately predict the consequences of interventions (e.g. because it misses a key dynamic that is not of great relevance in the absence of interventions). In such a case, one would then have done an analysis that conditions all future simulation on a single model that was only useful in the no-intervention setting.

This is another very valid point, and although we clarify that we focus on the no-intervention scenario, we now remark on this in the discussion (lines 4031-434).

In the end, I find that this is a very thorough analysis to illustrate a general approach to solving a model selection problem. My concern, however, is that it will be easy for some to interpret this as a final answer (e.g. use models with age, but not household, structure) rather than a protocol for how to answer the question about which type of model to use.

We thank the reviewer for both the praise and the concern, and we confess that addressing their concern has helped us clarifying to role of this work and where it fits in relation to other methods. We have extensively discussed and motivated this change of focus from the previous version.

Reviewer #2 (Remarks to the Author):

The current paper is concerned with model selection when analysing emerging epidemic outbreaks, and in particular studying what are the effects on prediction of choosing a simplified model.

This is an important topic and the paper contains some valuable conclusions (age-structure is more important than added within household transmission) based on an extensive simulation study. It is a welcome addition of an area not enough studied. I miss one paper which addresses the same type of question, but mainly from an analytical point of view: Trapman et al 2016 (Interface). I would have liked to see connections to conclusions there to see if they are in agreement or not (of course calibration is somewhat different).

We thank the reviewer for this comment. In fact, we had cited Trapman et al. 2016 (Interface) in the previous version of the manuscript, but due to page constraints and a less technical audience we were forced to relegate the citation to the supplementary material. We agree that this citation should appear in the main paper too, and we now extensively discuss it and its implication for our work in lines 214-217 and 346-355.

I think you can explain a bit better on p2-4 how the simulation study is done. You simulate according to AH model which results in R_0 , incidence ratio and SAR. Then, depending on the model used in statistical analysis you use 1-3 of these quantities to calibrate the model, and then compute final size, incidence peak and time of peak. At least this is how I interpreted it ...

The reviewer is correct. We have tried to rephrase the introduction to make this process clearer (lines 67-90, but see also lines 309-325).

All results are based on numerics/simulations. Of course it seems likely to be valid conclusions in relevant parameter regions, but still analytical results would of course strengthen this. They are hard to obtain, but perhaps worth mentioning in discussion.

We completely agree with the Reviewer on the importance of analytical results. However, we feel it is often hard to give a clear-cut definition of what one exactly means by “analytical”, so we prefer to distinguish “explicit” results from “analytical” ones (e.g. we would call the dominant eigenvalue of a positive matrix “analytical” even if it is not “explicit”). Explicit results in anything but the simplest Markovian model (that we do not use) are very rare. However, we still insist our approach is heavily based on analytical results, which is what makes that iterative methods used to solve the inverse problem in the mapping procedure efficient (simulations are only used for the “forward” computation of peak incidence and time to the peak, after the mapping has been performed). Nevertheless, the Reviewer is correct in pointing out that our conclusions are fundamentally numerical in nature, so we now clarify this in lines 309-325.

Minor comments

It is not clear to me why you study growth in terms of generations rather than in real time - the latter seems easier.

We are a bit unclear what the Reviewer is asking exactly here. We do not measure time in “generations”, but we use the “generation time” as a time unit (108-110). This is because we chose the case of influenza to fix some values for the time-dependent parameters we used, namely the mean and shape of the infectious contact interval distribution. This results, for examples, in predictions of the time to the peak that would be meaningful for influenza only. Rescaling such values by the generation time facilitates extrapolation to other infections.

If instead the reviewer is asking why we work with R_0 rather than the real-time growth rate r , we would reply that we had tried to briefly explain this point in the previous version of the manuscript, and a much deeper explanation was given in the SI. However, as it appears to be still

a potential source of confusion, the explanation in the main text has now been significantly expanded, to include also the discussion around Trapman et al. (2016). The main reason is that exact formulae for R_t in time-since-infection households models do not exist (though we have run Monte Carlo simulations in Sec. 8.2 of the SI to explore what would happen in the baseline scenario if R_t was used as observable). See lines 338-346.

p2-3: R_0 for household models is complicated (refer to discussion later in manuscript)
Done, thanks (lines 77-78).

Fig 1: The top row is the "truth"? If so this should be mentioned. Also, why no row for U model?
Done, thanks. We also clarified that results about model U would be constant and equal to the bottom-left corner of each panel, so are not plotted. See the caption of Figure 1.

p7, 159: "though by not much"?

We have replaced this with "can (just about) be ignored" (line 175). We meant that it can be ignored if the accuracy threshold is 5%, but we are very close to the border with the region where they can't be ignored. Knowing, as explained in lines 369-381, that the boundaries appear to change monotonically with most parameters, being close to the borders of regions suggests the relative discrepancy approaches the 5% threshold used.

p8, 227: systematically underestimates ...: I think this is in agreement with Trapman et al.
Yes, the Reviewer is correct, and we thank them for encouraging us to investigate this comparison further. We have extensively commented on it in lines 346-355.

p8, 229: then -> than
Corrected, thanks.

p9, middle: explain better what has been done mathematically
We have now significantly reworded this paragraph: see lines 309-325. We hope the new wording clarifies both what we have done and why analytical results are key to make our method efficient even if the conclusions are based on a numerical exploration of the parameter space.

p10, 296: that that
Corrected, thanks.

p10: 331-332: "as long as ... not affecting exponentially growing phase". Why would it not?
Good point. What we meant is that if observables are measured early enough that control policies are not implemented and spontaneous behavioural change is negligible, it would be meaningful to assume constant parameters (with values relative to the absence of intervention scenario) during the mapping procedure, and thus use all available analytical results (lines 426-431). Then, forward predictions could account for parameter changes due to control policies (assuming data availability), for which analytical results are unlikely to be available. However, that would be easy to incorporate, as the forward predictions, even with individual-based stochastic simulations, are relatively inexpensive compared to the mapping procedure.

p11: another thing not taken into account is spatial structure. This is of course harder, but still worth mentioning?
We have now added a remark about it (lines 448-450).

p11 bottom-12 top: first beta is expressed as a matrix (depending on age of infector + more) and then $R_0 = \beta$?
Thank you, we now call the function of time since infection $B(\tau)$, so that β only identifies its integral (lines 470-476).

p14+p15: mention that when analysing using Model A SAR is not used, and incidence ratio not used when using model H
Done, thanks (lines 632-633, 644-646 and 647-648).

REVIEWERS' COMMENTS:

Reviewer #1 (Remarks to the Author):

In general, I am satisfied with the authors' responses to my criticisms. I did notice the code access statement in this revision which states that code is "available from the authors upon request". Given that this is an entirely simulation-based analysis and, to my reading, there doesn't appear to be any proprietary data used in the manuscript, I think that it would be reasonable to require that the authors make all code available through an online repository. This would facilitate both review by readers and application of these methods as described by the authors.

Reviewer #2 (Remarks to the Author):

I am happy with the responses of the authors and hence support publication.

Response to Reviewers for manuscript NCOMMS-18-12337A, by Pellis, Cauchemez, Ferguson and Fraser.

Reviewer #1 (Remarks to the Author):

In general, I am satisfied with the authors' responses to my criticisms. I did notice the code access statement in this revision which states that code is "available from the authors upon request". Given that this is an entirely simulation-based analysis and, to my reading, there doesn't appear to be any proprietary data used in the manuscript, I think that it would be reasonable to require that the authors make all code available through an online repository. This would facilitate both review by readers and application of these methods as described by the authors.

The code is now available on GitHub, at <https://github.com/lorenzo-pellis/model-mapping>

Reviewer #2 (Remarks to the Author):

I am happy with the responses of the authors and hence support publication.

Thank you.